# β-catenin inhibition disrupts the homeostasis of osteogenic/adipogenic differentiation leading to the development of glucocorticoid-induced osteonecrosis of the femoral head

Chenjie Xia[1,2†], Huihui Xu[1,3†], Liang Fang[1], Jiali Chen[1], Wenhua Yuan[1], Danqing Fu[4], Xucheng Wang[1], Bangjian He[5], Luwei Xiao[1], Chengliang Wu[1], Peijian Tong[5], Di Chen[6*], Pinger Wang[1,3*], Hongting Jin[1,3*]

[1]Institute of Orthopedics and Traumatology, The First Affiliated Hospital of Zhejiang Chinese Medical University, Zhejiang Provincial Hospital of Chinese Medicine, Hangzhou, China; [2]Department of Orthopedic Surgery, the Affiliated Lihuili Hospital of Ningbo University, Ningbo, China; [3]The First College of Clinical Medicine, Zhejiang Chinese Medical University, Hangzhou, China; [4]School of Basic Medical Sciences, Zhejiang Chinese Medical University, Hangzhou, China; [5]Department of Orthopedic Surgery, the First Affiliated Hospital of Zhejiang Chinese Medical University, Hangzhou, China; [6]Faculty of Pharmaceutical Sciences, Shenzhen Institute of Advanced Technology, Shenzhen, China

*For correspondence:
di.chen@siat.ac.cn (DC);
pingerwang@zcmu.edu.cn (PW);
hongtingjin@163.com (HJ)

†These authors contributed equally to this work

Competing interest: The authors declare that no competing interests exist.

**Abstract** Glucocorticoid-induced osteonecrosis of the femoral head (GONFH) is a common refractory joint disease characterized by bone damage and the collapse of femoral head structure. However, the exact pathological mechanisms of GONFH remain unknown. Here, we observed abnormal osteogenesis and adipogenesis associated with decreased β-catenin in the necrotic femoral head of GONFH patients. In vivo and in vitro studies further revealed that glucocorticoid exposure disrupted osteogenic/adipogenic differentiation of bone marrow mesenchymal cells (BMSCs) by inhibiting β-catenin signaling in glucocorticoid-induced GONFH rats. Col2[+] lineage largely contributes to BMSCs and was found an osteogenic commitment in the femoral head through 9 mo of lineage trace. Specific deletion of β-catenin gene (*Ctnnb1*) in Col2[+] cells shifted their commitment from osteoblasts to adipocytes, leading to a full spectrum of disease phenotype of GONFH in adult mice. Overall, we uncover that β-catenin inhibition disrupting the homeostasis of osteogenic/adipogenic differentiation contributes to the development of GONFH and identify an ideal genetic-modified mouse model of GONFH.

## eLife assessment

This study presents **valuable** findings on the mechanism of glucocorticoid-induced osteonecrosis of the femoral head. The data were collected and analyzed using **solid**, validated methodology and can be used as a starting point for functional studies of development of glucocorticoid-induced osteonecrosis. This article would be of interest to cell biologists and biophysicists working on potential pharmacological treatments for glucocorticoid-induced osteonecrosis.

## Introduction

Glucocorticoids, potent immunity regulators, are widely used in the treatment of various autoimmune diseases such as systemic lupus erythematosus, nephrotic syndrome , and rheumatoid arthritis (*Yang et al., 2021*). However, excessive usage of glucocorticoids has been reported to cause severe adverse effects, especially femoral head osteonecrosis in the joint (*Zaidi et al., 2010*; *Tao et al., 2017*). As a destructive joint disease, glucocorticoid-induced osteonecrosis of the femoral head (GONFH) disables about 100,000 Chinese and more than 20,000 Americans annually (*Petek et al., 2019*; *Zhao et al., 2020*), bringing a huge financial burden on the society. Currently, there is no ideal medical treatment for GONFH due to its unclear pathological mechanisms. Most patients eventually have to undergo a joint replacement surgery (*Scaglione et al., 2015*). Therefore, in-depth study of GONFH pathogenesis is urgently needed.

Previous ex vivo studies on human necrotic femoral heads revealed prominent pathological features of GONFH, including fat droplet clusters, trabecular bone loss, empty lacunae of osteocyte at the early stage and extra subchondral bone destruction, and structure collapse at the late stage (*Qin et al., 2015*; *Wang et al., 2018a*). Glucocorticoid-induced animal models are commonly used to study GONFH pathogenesis, which mimic the early necrotic changes of GONFH but without structure collapse (*Zheng et al., 2018*). Several hypotheses, such as damaged blood supply (*Sun et al., 2021*; *Wang et al., 2018b*), abnormal lipid metabolism (*Lavernia et al., 1999*; *Chang et al., 1993*), and bone cell apoptosis (*Chen et al., 2022*; *Zalavras et al., 2003*), have been proposed to explain the occurrence and development of GONFH. However, the therapies developed according to the guidelines established based on these theories have not been successful in the prevention of GONFH progression. Clinical evidence about the decreased replication and osteogenic differentiation capacity of bone marrow mesenchymal stromal cells (BMSCs) (*Lee et al., 2006*; *Wang et al., 2008*; *Houdek et al., 2016*) and the efficacy of BMSC transplantation (*Wang et al., 2019b*; *Tabatabaee et al., 2015*; *Gangji and Hauzeur, 2005*) in GONFH patients indicate that GONFH might be a BMSC-related disease. BMSCs contain mesenchymal progenitor cells that give rise to osteoblasts and adipocytes, and glucocorticoids have been shown to regulate osteogenic/adipogenic differentiation of BMSCs in vitro (*Yin et al., 2006*; *Han et al., 2019*). Various treatments targeting promoting osteogenic differentiation of BMSCs alleviate early necrotic phenotype in glucocorticoid-induced GONFH animal models (*Chen et al., 2020*; *Zhu et al., 2020*; *Feng et al., 2022*). Thus, we hypothesize that imbalanced osteogenic/adipogenic differentiation of BMSCs plays a dominant role in GONFH pathogenesis. Recent lineage-tracing and single-cell RNA-sequencing studies revealed that BMSCs are a group of heterogeneous multipotent cells (*Gao et al., 2021*), and skeletal-derived mesenchymal progenitor cells, including collagen II (Col2)[+] lineage and Osterix (Sp7)[+] lineage, compose a large proportion of BMSCs (*Ono et al., 2014*; *Liu et al., 2013*). However, which subpopulation of BMSCs significantly contributes to the GONFH pathogenesis remains unknown.

BMSC differentiation is a complex process and is regulated by multiple signaling pathways. Among them, canonical Wnt/β-catenin signaling functions as a switch in determining osteogenic/adipogenic differentiation of BMSCs (*Song et al., 2012*). When β-catenin is accumulated in the nucleus, it interacts with TCF/Lef transcription factors to activate downstream target genes, including *Runt-related transcription factor 2* (*Runx2*) and *Sp7* (*Yuan et al., 2016*). On the contrary, inhibition of β-catenin induces expressions of *CCAAT/enhancer binding protein alpha* (*Cebpa*) and *peroxisome proliferator-activated receptor gamma* (*Pparg*) for adipogenesis (*Takada et al., 2009*; *Xu et al., 2016*). Our previous study showed a significant decrease in β-catenin in the glucocorticoid-induced GONFH rat model (*Zhang et al., 2019*). Other groups also reported an involvement of β-catenin signaling in GONFH development (*Zhang et al., 2021*; *Chen et al., 2019*; *Zhao et al., 2021*; *Yu et al., 2016*; *Zhang et al., 2015*).

Here, abnormal osteogenesis and adipogenesis with decreased β-catenin signaling were observed in the necrotic femoral heads of GONFH patients and glucocorticoid-induced GONFH rats. Activation of β-catenin signaling effectively alleviated the necrotic changes in GONFH rats by restoring glucocorticoid exposure-induced imbalanced osteogenic/adipogenic differentiation of BMSCs. Interestingly, specific deletion of β-catenin gene (*Ctnnb1*) in Col2[+] cells, but not Sp7[+], cells led to a full spectrum of disease phenotype of GONFH in mice even including subchondral bone destruction and femoral head collapse. This study provides novel molecular mechanisms of GONFH pathogenesis and an ideal genetic-modified mouse model for GONFH study.

## Results

### Abnormal osteogenesis and adipogenesis with decreased β-catenin in necrotic femoral heads of GONFH patients

To determine the pathogenesis of GONFH, we harvested the surgical specimens from GONFH patients with femoral head necrosis (n = 15) and samples from trauma patients without femoral head necrosis (n = 10). Compared to the non-necrotic femoral heads, we observed subchondral bone destruction (*Figure 1A*, black asterisk), liquefied necrotic foci (*Figure 1A*, black arrow), and deformed and collapsed outline in the necrotic femoral head. μCT analysis further showed sparse and cracked trabeculae in the collapsed region (*Figure 1B*, red box) and liquefied necrotic region (*Figure 1B*, yellow box) compared to the corresponding regions (*Figure 1B*, white boxes) in the non-necrotic femoral heads, which were further confirmed by the decreased BV/TV, Tb.N, and Tb.Th and increased Tb.Sp (*Figure 1C*). The representative images of Alcian Blue Hematoxylin (ABH) staining (*Figure 1D*) revealed sparse trabeculae, massive accumulated fat droplets (*Figure 1D*, black triangles), numerous empty lacunae of osteocytes (*Figure 1D*, black arrows), and extensive subchondral bone destruction (*Figure 1D*, yellow arrows) in the necrotic femoral heads. Histomorphological quantitative analyses showed decreased trabecular bone area (*Figure 1E*), increased fat droplet area (*Figure 1E*), and empty lacunae rate (*Figure 1F*) in the necrotic femoral heads. TUNEL staining revealed more apoptotic osteocytes in the necrotic femoral heads (*Figure 1G*). We then detected the expression of osteogenic (*Runx2*, *Sp7*) and adipogenic (*Fabp4*, *Pparg*, *Cebpa*) marker genes. The results of immunohistochemistry (IHC) staining showed a downregulation of Runx2 and an upregulation of FABP4 in the necrotic regions (*Figure 1H*). Western blot analysis showed increased expressions of PPAR-γ and C/EBP-α and a decreased expression of Osterix in the human necrotic femoral head tissues (*Figure 1I*). These findings indicate an involvement of abnormal osteogenesis and adipogenesis during GONFH pathogenesis. β-Catenin can regulate osteogenesis and adipogenesis downstream target genes (*Runx2*, *Sp7*, *Fabp4*, *Pparg*, and *Cebpa*) expressions (*Tencerova and Kassem, 2016*). The significant decrease in β-catenin in the necrotic femoral heads (*Figure 1G and H*) indicates an important role of β-catenin signaling in GONFH pathogenesis.

### Inhibition of β-catenin signaling leads to abnormal osteogenesis and adipogenesis in glucocorticoid-induced GONFH rats

To further analyze the role of β-catenin in GONFH pathogenesis, a glucocorticoid-induced GONFH rat model was established by continuous methylprednisolone (MPS) induction, as previously described (*Zheng et al., 2018*). Gross anatomy analysis showed a local melanocratic region on the femoral head surface of GONFH rats without any structure deformity or collapse (*Figure 2A*, black arrow). No color change was observed on the femoral head surface from the Wnt agonist 1-treated rats. μCT images and quantitative analysis of bone microstructure showed severe trabecular bone loss in the femoral heads of GONFH rats, especially in the subchondral region (*Figure 2A and B*). Systemic injection of Wnt agonist 1 effectively increased bone mass (*Figure 2A*), and alleviated decreased BV/TV, Tb.N, and Tb.Th and increased Tb.Sp in GONFH rats (*Figure 2B*). ABH staining and histomorphological quantitative analysis revealed that the GONFH rats presented sparse trabeculae, massive fat accumulation (*Figure 2C*, black arrow), and numerous empty lacunae of osteocytes (*Figure 2C*, black arrows) in the necrotic region of femoral heads (*Figure 2C–E*). However, no subchondral bone destruction, femoral head collapse, and deformity occurred in this GONFH rat model (*Figure 2C*). TUNEL staining revealed increased apoptotic osteocytes in the femoral heads of GONFH rats (*Figure 2F*). ALP is an osteoblast marker reflecting the capacity of osteogenesis. FABP4 is essential for lipid formation and metabolism. The decreased β-catenin and ALP expressions and increased FABP4 expression confirmed abnormal osteogenesis and adipogenesis during rat GONFH development (*Figure 2G and H*), consistent with the findings in human GONFH. Interestingly, activation of β-catenin by systematically injecting Wnt agonist 1 for 6 wk attenuated the early necrotic changes (*Figure 2C–F*), and restored ALP and FABP4 expressions in the femoral heads of GONFH rats (*Figure 2G and H*). Overall, these data indicate that β-catenin inhibition-induced abnormal osteogenesis and adipogenesis contributes to GONFH pathogenesis.

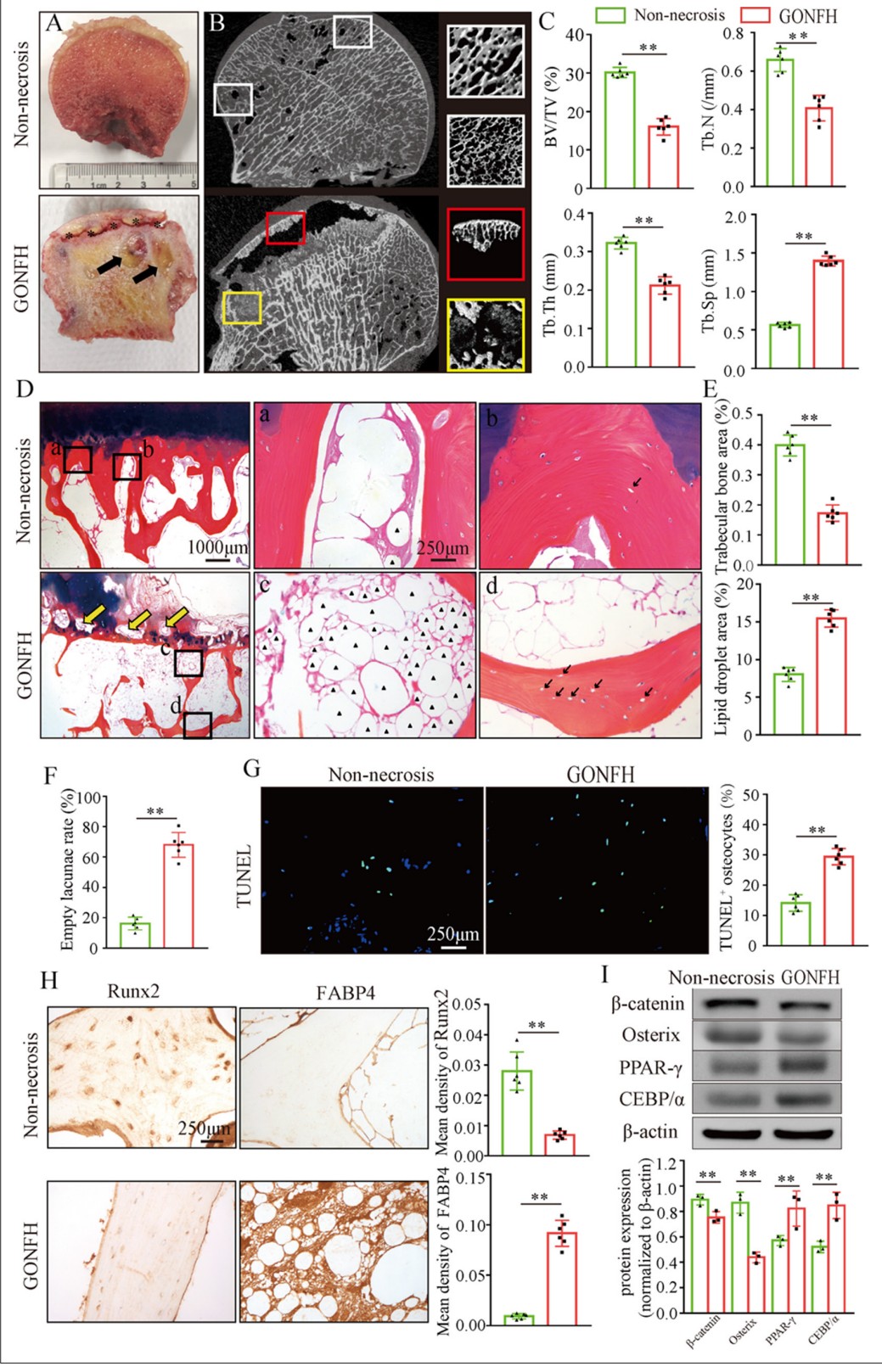

**Figure 1.** Abnormal osteogenesis and adipogenesis with decreased β-catenin signaling in the necrotic femoral heads of glucocorticoid-induced osteonecrosis of the femoral head (GONFH) patients. Necrotic (n = 15) and non-necrotic (n = 10) femoral head samples were obtained from GONFH patients or femoral neck fracture patients, respectively. (**A**) Gross anatomy analysis of human necrotic and non-necrotic femoral head samples. Black

*Figure 1 continued on next page*

*Figure 1 continued*

asterisks: subchondral collapsed region. Black arrows: liquefied necrotic region. (**B**) µCT images of human necrotic and non-necrotic femoral heads. Red and yellow boxed areas: 3D images of subchondral collapsed region and liquefied necrotic region, respectively. White boxed areas: 3D images of corresponding regions in the non-necrotic femoral heads. (**C**) Quantitative analysis of BV/TV, Tb.N, Tb.Th, and Tb.Sp on the necrotic regions. (**D**) Alcian Blue Hematoxylin (ABH) staining of human necrotic and non-necrotic femoral heads. (a, c) High-magnification images of bone marrow; (b, d) high-magnification images of bone trabeculae; yellow arrows: subchondral bone destruction; black arrows: empty lacunae of osteocytes; black triangles: fat droplets. (**E**) Histomorphological quantitative analysis of trabecular bone area and fat droplet area. (**F**) Histomorphological quantitative analysis of empty lacunae rate. (**G**) TUNEL staining of osteocytes in human necrotic and non-necrotic femoral heads. (**H**) Immunohistochemistry (IHC) staining of Runx2 and FABP4 expressions in human necrotic and non-necrotic femoral heads. (**I**) Western blot of β-catenin, Osterix, PPAR-γ, and CEBP/α in human necrotic and non-necrotic femoral heads.

The online version of this article includes the following source data for figure 1:

**Source data 1.** Raw data for *Figure 1*.

**Source data 2.** Labeled uncropped western blots for *Figure 1*.

**Source data 3.** Raw unedited blots for *Figure 1*.

## β-catenin directs dexamethasone-induced osteogenic/adipogenic differentiation of rat BMSCs

We further investigated cellular mechanisms involved in the necrotic femoral heads of glucocorticoid-induced GONFH rats. Primary bone marrow cells were extracted from the proximal femur of 4-week-old Sprague–Dawley (SD) rats and cultivated to P3. Flow cytometry analysis identified that they were BMSCs, with CD90+, CD29+, CD45-, and CD11b- characteristics (*Figure 3A*). ALP staining showed that the osteogenic differentiation of BMSCs was significantly inhibited after exposing to $10^2$ nM or higher concentration of dexamethasone (Dex) for 7 d (*Figure 3B*). Oil red O staining showed a progressive adipogenic differentiation of BMSCs with the increased concentration of Dex (*Figure 3B*). These findings indicated that increased adipogenic differentiation and decreased osteogenic differentiation of BMSCs caused the abnormal osteogenesis and adipogenesis, finally leading to the femoral head necrosis in GONFH rats. The decreased expressions of β-catenin, Runx2, ALP, and increased expressions of PPAR-γ, CEBP/α further revealed that Dex exposure inhibited β-catenin signaling and redirected BMSCs differentiation from osteoblasts to adipocytes (*Figure 3C and D*). However, this Dex-induced abnormal differentiation of BMSCs was restored by SKL2001 through activation of β-catenin signaling (*Figure 3E–G*). Taken together, these findings suggest that β-catenin inhibition redirects the direction of BMSC differentiation, contributing to the GONFH development.

## Loss of function of *Ctnnb1* in Col2+-expressing, but not Sp7+, cells leads to a GONFH-like phenotype in femoral head

Skeletal progenitor cells are the major source of BMSCs, and both Col2+ lineage and Sp7+ lineage can trans-differentiate into BMSCs (*Gao et al., 2021*; *Ono et al., 2014*). To determine which subset of BMSCs contributes more to GONFH pathogenesis, we generated *Col2a1-CreER^T2^;Ctnnb1^flox/flox^* (Ctnnb1^Col2ER^) mice and *Sp7-CreER^T2^;Ctnnb1^flox/flox^* (Ctnnb1^Sp7ER^) mice, in which *Ctnnb1* gene was specifically deleted in Col2+ cells and Sp7+ cells, respectively. For mapping their cell fate in the femoral head, we induced *Col2a1-CreER^T2^;Rosa26-LSL-tdTomato* (Tomato^Col2ER^) mice and *Sp7-CreER^T2^;Rosa26-LSL-tdTomato* (Tomato^Sp7ER^) mice at 2 weeks old and analyzed at the age of 1 mo. The fluorescent images showed that Col2+ cells were evidently expressed in the chondrocytes of articular cartilage, secondary ossification center and growth plate (*Figure 4A*, white arrowheads), osteoblasts (*Figure 4A*, green arrowheads), osteocytes (*Figure 4A*, yellow arrowheads), and stromal cells (*Figure 4A*, red arrowheads) underneath the growth plate, while Sp7+ cells were identified only in osteoblasts, osteocytes, and stromal cells underneath the growth plate (*Figure 4A*). We then trace Col2+ lineage for 9 mo and found that Col2+ chondrocytes in secondary ossification center and growth plate were replaced by Col2+ osteoblasts, osteocytes, and stromal cells with age (*Figure 4B*), and these Col2+ cells continuously produced osteoblasts and osteocytes in the femoral head (*Figure 4B*). These findings indicated the self-renew ability and osteogenic commitment of Col2+ cells. We also induced

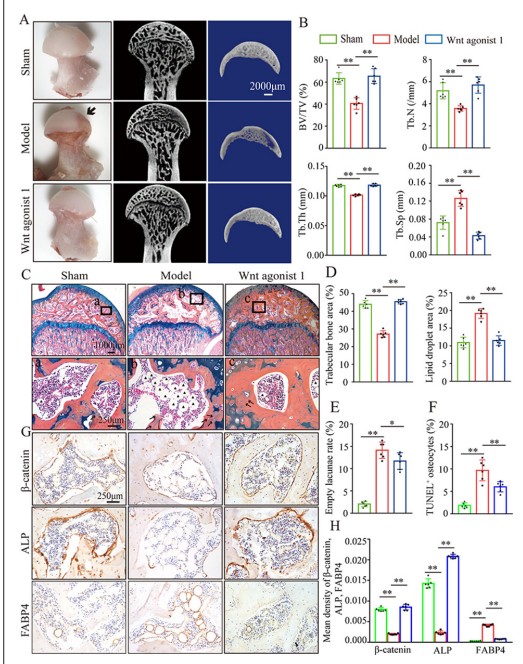

**Figure 2.** Systemic injection of Wnt agonist 1 alleviates abnormal osteogenesis and adipogenesis in rat glucocorticoid-induced osteonecrosis of the femoral head (GONFH). The rat model of GONFH was established by a co-induction of lipopolysaccharide and methylprednisolone (MPS). The femoral head samples were harvested after GONFH rats were intravenously injected with Wnt agonist 1 for 6 wk. (**A**) Gross anatomy analysis and μCT images of femoral heads in sham, model, and Wnt agonist 1-treated groups. (**B**) μCT quantitative analysis of BV/TV, Tb.N, Tb.Th, and Tb.Sp in each group. (**C**) Alcian Blue Hematoxylin (ABH) staining of femoral heads in each group. (a–c) High-magnification images of the representative region; black arrows: empty lacunae of osteocytes; black triangles: fat droplets. (**D**) Histomorphological quantitative analysis of trabecular bone area and fat droplet area in each group. (**E**) Histomorphological quantitative analysis of empty lacunae rate in each group. (**F**) TUNEL staining of osteocytes in the femoral heads of GONFH rats. (**G, H**) Immunohistochemistry (IHC) staining and quantitative analysis of β-catenin, ALP, and FABP4 in rat femoral heads of each group.

The online version of this article includes the following source data for figure 2:

**Source data 1.** Raw data for **Figure 2**.

TomatoCol2ER mice at the age of 1 mo, but identified a poor *Cre*-recombinase efficiency 24 hr and 3 mo later (**Figure 4C**). IHC assay showed a significant decrease of β-catenin in the femoral heads of 3-month-old Ctnnb1Col2ER mice and Ctnnb1Sp7ER mice (**Figure 4D and E**), indicating successful establishments of these *Ctnnb1* conditional knockout mouse models.

Then, we analyzed morphological changes of the femoral heads from 3-month-old Ctnnb1Col2ER mice and Ctnnb1Sp7ER mice. No change of appearance was observed on the femoral heads both in Ctnnb1Col2ER mice and Ctnnb1Sp7ER mice compared to *Cre*-negative littermates (**Figures 5A and 6A**). μCT images and quantitative analysis of decreased BV/TV, Tb.N, Tb.Th, and increased Tb.Sp indicated that both Ctnnb1Col2ER mice and Ctnnb1Sp7ER mice presented severe bone loss in the femoral heads (**Figures 5B and 6B**). ABH staining and histomorphological analysis further revealed that bone trabeculae were extensively replaced by fat droplet clusters in the femoral heads (especially in the subchondral region) of Ctnnb1Col2ER mice (**Figure 5C and D**), similar to the necrotic femoral heads in GONFH patients and glucocorticoid-induced rat models. However, no fat droplet accumulation but only sparse and thin bone trabeculae were observed in the femoral heads of Ctnnb1Sp7ER mice (**Figure 6C and D**). Compared to *Cre*-negative littermates, the number of empty lacunae was increased in the femoral heads in Ctnnb1Col2ER mice (**Figure 5E**), but not changed in Ctnnb1Sp7ER mice (**Figure 6E**). TUNEL staining revealed more apoptotic osteocytes in Ctnnb1Col2ER mice compared to *Cre*-negative littermates (**Figure 5F**). All these findings indicate that loss of function of *Ctnnb1* in Col2+ cells, but not in Sp7+ cells, led to a GONFH-like phenotype in femoral head. Furthermore, decreased Runx2, ALP and increased PPAR-γ, CEBP/α protein expressions in Ctnnb1Col2ER mice (**Figure 5G**) revealed that this GONFH-like phenotype was caused by imbalance of osteogenic/adipogenic differentiation of Col2+ cells.

## Older Ctnnb1Col2ER mice display subchondral bone destruction and collapse tendency in the femoral heads

Femoral head collapse occurs at the end stage of GONFH and represents a poor joint function (*Hines et al., 2021*). To determine the occurrence of femoral head collapse, we further analyzed 6-month-old Ctnnb1Col2ER mice. Gross appearance showed that older Ctnnb1Col2ER mice lost smooth and regular outline on the femoral heads compared to *Cre*-negative littermates and 3-month-old Ctnnb1Col2ER mice (**Figure 7A**). μCT images showed severe bone loss (**Figure 7B and C**), subchondral bone destruction (**Figure 7B**, yellow arrows), local collapse (**Figure 7B**, red arrows), and integral deformity (**Figure 7B**,

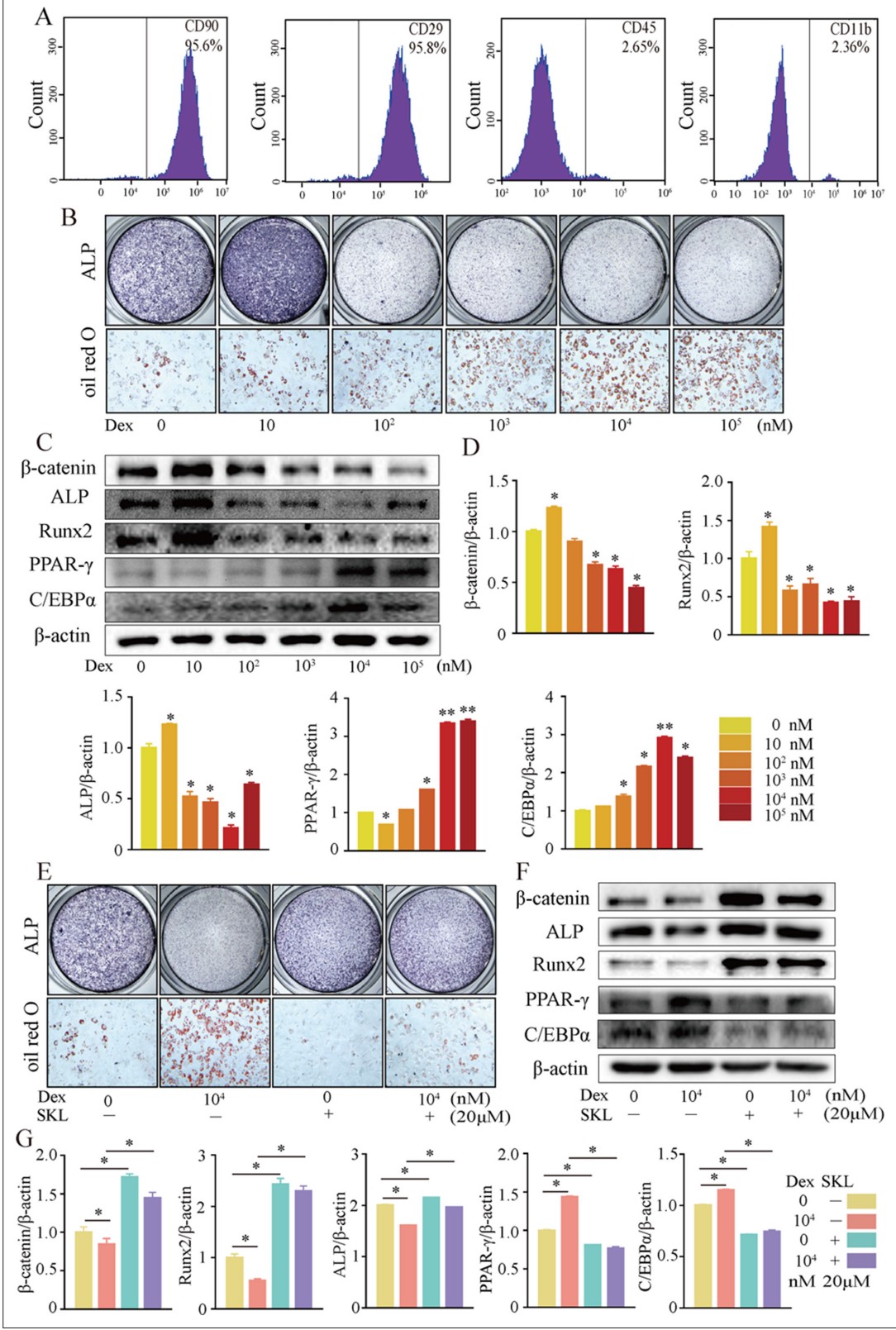

**Figure 3.** Activation of β-catenin redirected dexamethasone (Dex)-induced imbalanced osteogenic/adipogenic differentiation of bone marrow mesenchymal cells (BMSCs). Primary BMSCs were extracted from the proximal femur of 4-week-old Sprague–Dawley (SD) rats and cultivated to passage 3 for subsequent experiments. (**A**) Flow cytometry analyzing the surface markers of rat BMSCs. (**B**) ALP staining and oil red O staining of BMSCs at the

*Figure 3 continued on next page*

*Figure 3 continued*

increasing concentrations of Dex. (**C, D**) Western blot and quantitative analysis of β-catenin, Runx2, ALP, PPAR-γ, and CEBP/α in BMSCs at the increasing concentrations of Dex. (**E**) ALP staining and oil red O staining of rat BMSCs at the condition with or without $10^4$ nM Dex and 20 μM SKL2001 (an agonist of β-catenin). (**F, G**) Western blot and quantitative analysis of β-catenin, Runx2, ALP, PPAR-γ, and CEBP/α in rat BMSCs at the condition with or without $10^4$ nM Dex and 20 μM SKL2001.

The online version of this article includes the following source data for figure 3:

**Source data 1.** Raw data for *Figure 3*.

**Source data 2.** Labeled uncropped western blots for *Figure 3*.

**Source data 3.** Raw unedited blots for *Figure 3*.

---

red dotted lines) in the femoral heads of older Ctnnb1[Col2ER] mice. μCT quantitative analysis confirmed a significant increase of subchondral bone defect area on the femoral head surface in older Ctnnb1[Col2ER] mice (*Figure 7D*). Besides these early necrotic changes including sparse bone trabeculae, accumulated fat droplets (*Figure 7E*, black triangles), and increased empty lacunae of osteoctyes (*Figure 7E*, black arrows, *Figure 7F*), ABH staining also showed subchondral bone destruction (*Figure 7*, yellow arrows), and femoral head collapse or deformity (*Figure 7E*, red arrows and red dotted lines) in older Ctnnb1[Col2ER] mice. Compared to *Cre*-negative littermates, the older Ctnnb1[Col2ER] mice presented a decrease of loading-bearing stiffness, indicating a poor biomechanical support of the femoral heads (*Figure 7G*).

## Wnt agonist 1 could not alleviate GONFH-like phenotype in Ctnnb1[Col2ER] mice

Regarding the therapeutic effects of Wnt agonist 1 on rat GONFH, we further determined whether it could alleviate the GONFH-like phenotype in Ctnnb1[Col2ER] mice. After induction with TM, Ctnnb1[Col2ER] mice were immediately injected with Wnt agonist 1 three times once a week until they were analyzed at the age of 3 mo. However, μCT images and quantitative analysis of BV/TV, Tb.N, Tb.Th, and Tb.Sp showed that severe bone loss was not restored in Ctnnb1[Col2ER] mice by treating with Wnt agonist 1 (*Figure 8A and B*). Accumulated fat droplets, sparse trabeculae, and empty lacunae still existed in the femoral heads of Wnt agonist 1-treated Ctnnb1[Col2ER] mice (*Figure 8C–E*). IHC staining showed no change of decreased ALP and increased FABP4 in the femoral heads of Ctnnb1[Col2ER] mice after Wnt agonist 1 treatment (*Figure 8F and G*). These findings indicate that Wnt agonist 1 could not restore the imbalanced osteogenic/adipogenic differentiation of Col2[+] cells in Ctnnb1[Col2ER] mice.

## Discussion

Despite being well known that glucocorticoids are the certain etiology of GONFH, its pathogenesis remains largely unknown. Current evidence, including reduced osteogenic differentiation ability of BMSCs from GONFH patients and clinical therapeutical effects of BMSCs transplantation, indicates that GONFH could result from abnormal differentiation of BMSCs (*Houdek et al., 2016*; *Wang et al., 2019b*; *Tabatabaee et al., 2015*; *Gangji and Hauzeur, 2005*). Glucocorticoid-induced animal models are commonly used for GONFH research, and several studies have revealed that direct BMSCs modifications with miRNA (*Cao et al., 2021*; *Gu et al., 2016*), lncRNA (*Wang et al., 2018c*; *Xu et al., 2022*), and circRNA (*Chen et al., 2020*; *Xiang et al., 2020*) or microenvironment interventions by injecting platelet-rich plasma (*Xu et al., 2021*; *Wang et al., 2022*), exosome (*Li et al., 2020*; *Fang et al., 2019*) and growth factors (*Guzman et al., 2021*; *Rackwitz et al., 2012*) could promote osteogenic differentiation of BMSCs and exhibit anti-GONFH effects in animals. In the present study, we found severe trabecular bone loss and massive fat droplet accumulation in the necrotic femoral heads of GONFH patients and glucocorticoid-induced GONFH rats, confirming increased adipogenesis and decreased osteogenesis in GONFH pathogenesis. Dex inhibited osteogenic differentiation of rat BMSCs and promoted their adipogenic differentiation, revealing that this abnormal osteogenesis and adipogenesis in GONFH rats were caused by the imbalance in osteogenic/adipogenic differentiation of BMSCs. As β-catenin plays an important role in BMSCs differentiation, we then analyzed its role in GONFH pathogenesis. The results showed a significant inhibition of β-catenin signaling in the necrotic femoral

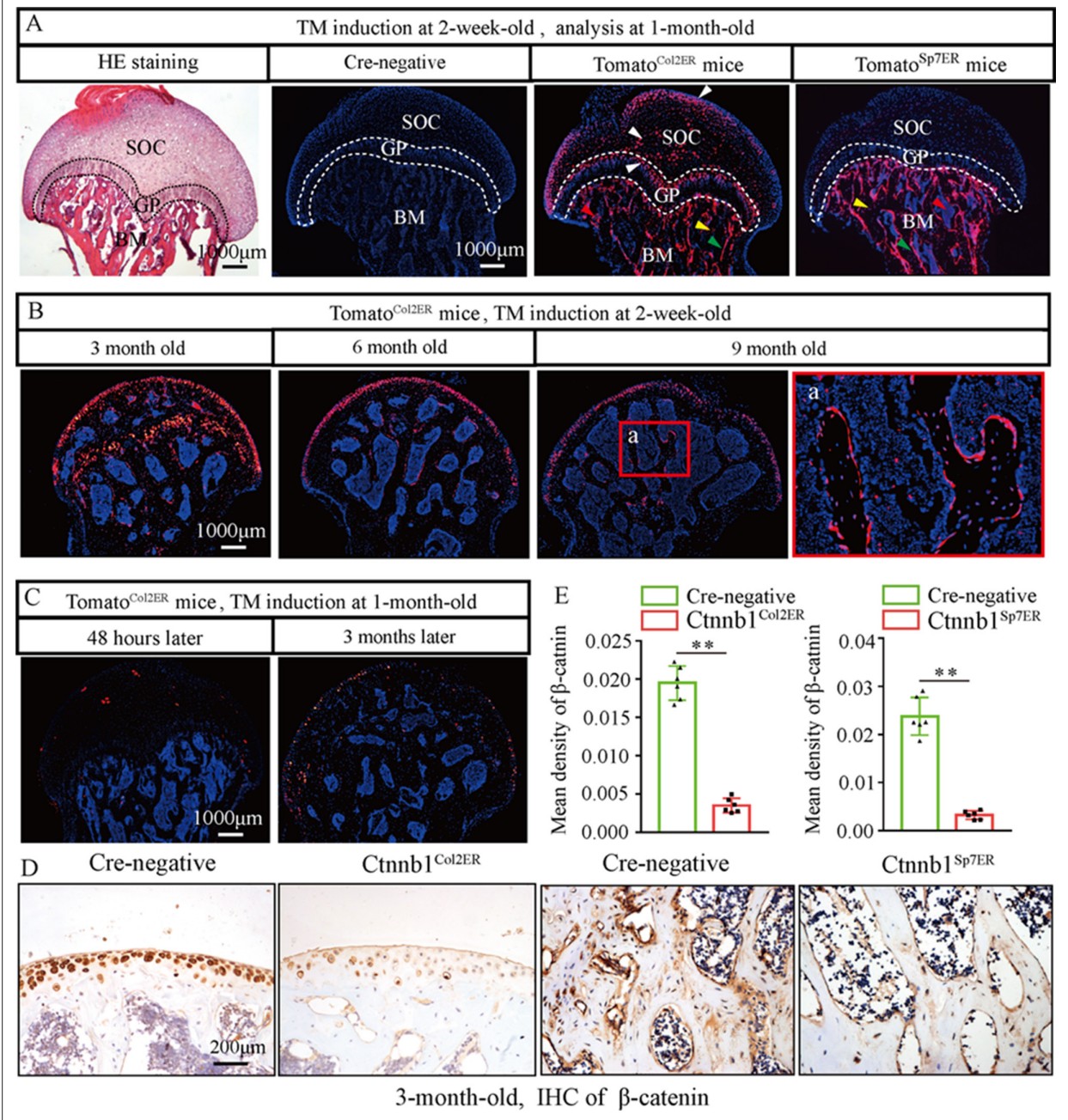

**Figure 4.** Fate mapping of Col2+ cells and Sp7+ cells in the femoral head. Tomato<sup>Col2ER</sup> mice and Tomato<sup>Sp7ER</sup> mice continuously received five doses of tamoxifen (TM) injections (1 mg/10 g body weight) at the age of 2 wk for fate mapping analysis. (**A**) Distributions of Col2+ and Sp7+ cells in the femoral heads of 1-month-old Tomato<sup>Col2ER</sup> mice and Tomato<sup>Sp7ER</sup> mice. White arrowheads: chondrocytes; green arrowheads: osteoblasts; yellow arrowheads: osteocytes; red arrowheads: bone marrow stromal cells; GP: growth plate; SOC: second ossification center; BM: bone marrow. (**B**) Lineage trace of Col2+ cells in the femoral head for 9 mo. a: high-magnification image of bone marrow region. (**C**) Poor *Cre*-recombinase efficiency in the femoral heads of Tomato<sup>Col2ER</sup> mice with TM induction at the age of 1 mo. (**D, E**) Femoral heads were harvested from 3-month-old Ctnnb1<sup>Col2ER</sup> mice and Ctnnb1<sup>Sp7ER</sup> mice to detect expression of β-catenin. Immunohistochemistry (IHC) staining and quantitative analysis of β-catenin in the femoral heads of 3-month-old Ctnnb1<sup>Col2ER</sup> mice and Ctnnb1<sup>Sp7ER</sup> mice.

The online version of this article includes the following source data for figure 4:

**Source data 1.** Raw data for *Figure 4*.

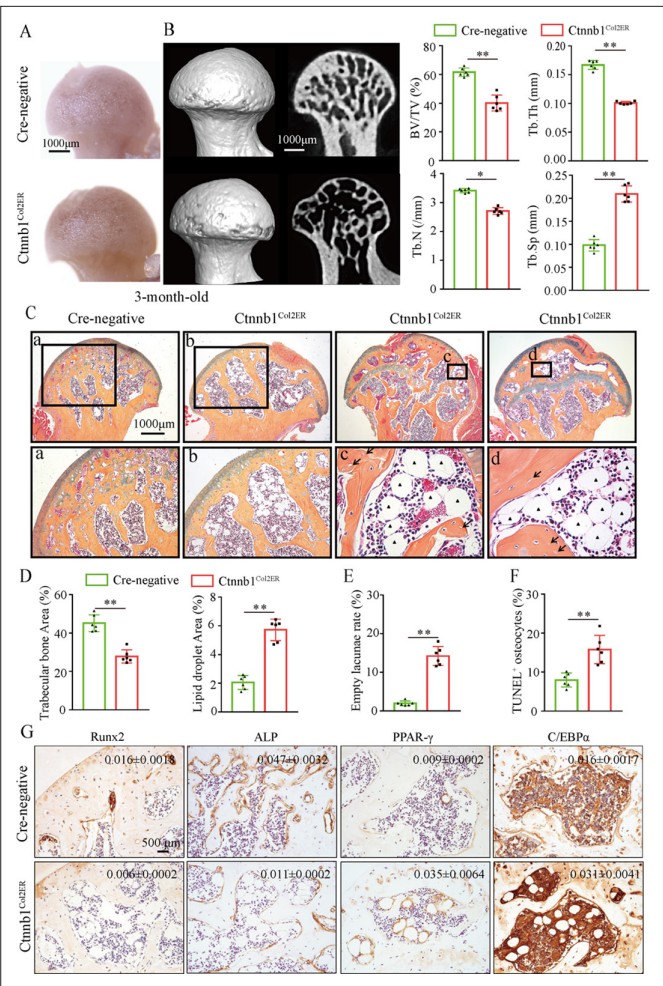

**Figure 5.** Deletion of *Ctnnb1* in Col2[+] cells leads to a glucocorticoid-induced osteonecrosis of the femoral head (GONFH)-like phenotype. Femoral heads were harvested from 3-month-old Ctnnb1[Col2ER] mice and *Cre*-negative littermates with five continuous dosages of tamoxifen (TM) injections (1 mg/10 g body weight) at the age of 2 wk. (**A**) Gross anatomy analysis of femoral heads in Ctnnb1[Col2ER] mice and *Cre*-negative littermates. (**B**) Representative μCT images and quantitative analysis of BV/TV, Tb.N, Tb.Th, and Tb.Sp in the femoral heads of Ctnnb1[Col2ER] mice. (**C**) Alcian Blue Hematoxylin (ABH) staining of femoral heads in Ctnnb1[Col2ER] mice. (a–d) High-magnification images of representative subchondral bone region; black triangle arrowheads: fat droplets; black arrows: empty lacunae of osteocytes. (**D**) Histomorphological quantitative analysis of trabecular bone area and fat droplet area in the femoral heads of Ctnnb1[Col2ER] mice. (**E**) Histomorphological quantitative analysis of empty lacunae rate in the femoral heads of Ctnnb1[Col2ER] mice. (**F**) TUNEL staining of osteocytes in the femoral heads of Ctnnb1[Col2ER] mice. (**G**) Immunohistochemistry (IHC) staining and quantitative analysis of Runx2, ALP, PPAR-γ, and CEBP/α in the femoral heads of Ctnnb1[Col2ER] mice.

The online version of this article includes the following source data for figure 5:

**Source data 1.** Raw data for *Figure 5*.

heads of GONFH patients and glucocorticoid-induced GONFH rats, while Wnt agonist 1 attenuated accumulated fat droplets and sparse trabeculae in GONFH rats by activation of β-catenin signaling. The subsequent cellular experiments revealed that β-catenin activation restored the imbalance of osteogenic/adipogenic differentiation of BMSCs caused by long-term exposure to Dex. Overall, these findings indicate that β-catenin inhibition-induced imbalanced osteogenic/adipogenic differentiation of BMSCs contributed to GONFH.

BMSCs are a group of heterogeneous cells, and the majority are derived from skeletal progenitors (*Gao et al., 2021*). It has reported that both Col2[+] progenitors and Sp7[+] progenitors provide a large proportion of BMSCs (*Ono et al., 2014*). Adult transgenic mice with conditional deletion of

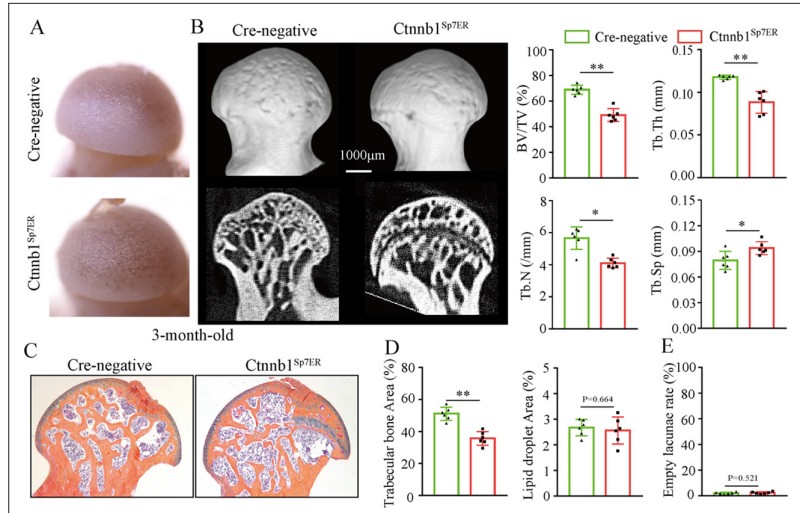

**Figure 6.** Loss of function of *Ctnnb1* in Sp7[+] cells causes bone loss in the femoral heads. Femoral heads were harvested from 3-month-old Ctnnb1[Sp7ER] mice and *Cre*-negative littermates with five continuous dosages of tamoxifen (TM) injections (1 mg/10 g body weight) at the age of 2 wk. (**A**) Gross anatomy analysis of femoral heads in Ctnnb1[Sp7ER] mice and *Cre*-negative littermates. (**B**) µCT images and quantitative analysis of BV/TV, Tb.N, Tb.Th, and Tb.Sp in the femoral heads of Ctnnb1[Sp7ER] mice. (**C**) Alcian Blue Hematoxylin (ABH) staining of femoral heads in Ctnnb1[Sp7ER] mice. (**D**) Histomorphological quantitative analysis of trabecular bone area and fat droplet area in the femoral heads of Ctnnb1[Sp7ER] mice. (**E**) Histomorphological quantitative analysis of empty lacunae rate in the femoral heads of Ctnnb1[Sp7ER] mice.

The online version of this article includes the following source data for figure 6:

**Source data 1.** Raw data for *Figure 6*.

---

*Ctnnb1* in Col2[+] cells, but not Sp7[+] cells, displayed a GONFH-like phenotype. Extensive trabecular bone was replaced by fat droplets in the femoral head (especially in the subchondral bone region) of Ctnnb1[Col2ER] mice compared to Ctnnb1[Sp7ER] mice. Cell fate mapping revealed an apparent difference in the distributions of Sp7[+] cells and Col2[+] cells in the femoral head when mice were induced at 2 weeks old. Besides the similar parts with Sp7[+] cells (osteoblasts, osteocytes, and stromal cells underneath growth plate), Col2[+] cells are also largely expressed in the chondrocytes of articular cartilage, secondary ossification center, and growth plate. Whether these Col2[+] chondrocytes in secondary ossification center and growth plate caused this GONFH-like phenotype, the osteogenic commitment of these Col2[+] chondrocytes has been reported (*Shu et al., 2021*; *Li et al., 2022*). At the age of 3 mo, Col2[+] chondrocytes in the secondary ossification center and growth plate were almost replaced by Col2[+] osteoblasts, osteocytes, and stromal cells, while GONFH-like phenotype, including fat droplet accumulation and sparse bone trabeculae, could be still observed in 6-month-old Ctnnb1[Col2ER] mice, indicating that not only Col2[+] chondrocytes but all Col2[+] cells in different cellular morphology were involved in this GONFH-like phenotype. Nine months of lineage trace further revealed that Col2[+] cells had self-renew ability and could continuously produce osteoblasts. The decreased ALP, Runx2 and increased CEBP/α and PPAR-γ in Ctnnb1[Col2ER] mice revealed that inhibition of β-catenin shifted the commitment of Col2[+] cells from osteoblasts to adipocytes, which should be responsible for this GONFH-like phenotype.

Femoral head collapse is the key feature of late-stage GONFH and is disastrous to hip joint (*Wang et al., 2019a*). Although it has been well recognized that poor mechanical support caused structural collapse of necrotic femoral head (*Fan et al., 2011*), its detailed pathogenesis is poorly understood due to the lack of ideal animal models. Glucocorticoid-induced bipedal and quadrupedal animal models only exhibit the early necrotic changes of GONFH (*Xu et al., 2018*), without structure collapse of femoral head in MPS-induced rats. Other models using chemical or physical approaches all failed to mimic full-range osteonecrosis of GONFH in humans (*Wang et al., 2021*). Importantly, we found that femoral head collapse occurred spontaneously in older Ctnnb1[Col2ER] mice. Besides the early necrotic changes, including trabecular bone loss, empty lacunae, and fat droplet accumulation, extensive

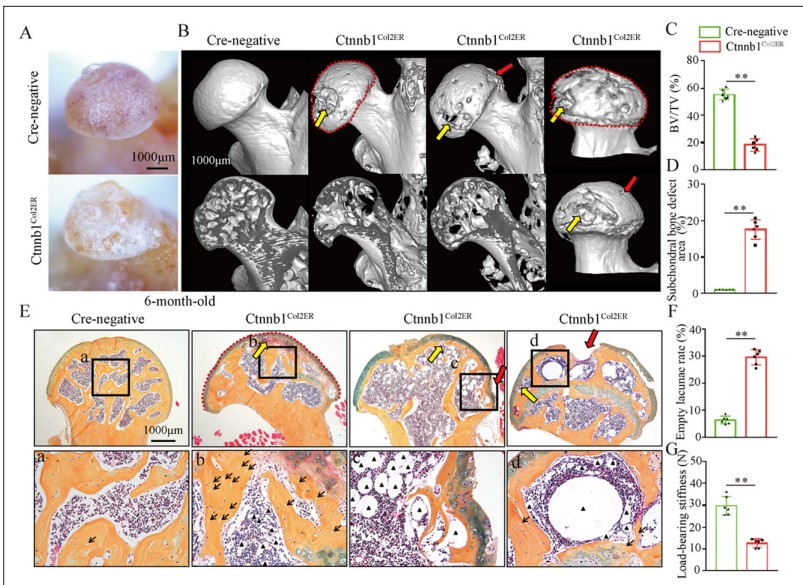

**Figure 7.** Older Ctnnb1[Col2ER] mice display subchondral bone destruction and collapse tendency in femoral heads. Femoral heads were harvested from 6-month-old Ctnnb1[Col2ER] mice and *Cre*-negative littermates with five continuous dosages of tamoxifen (TM) injections (1 mg/10 g body weight) at the age of 2 wk. (**A**) Gross anatomy analysis of femoral heads in 6-month-old Ctnnb1[Col2ER] mice. (**B**) Representative µCT images of femoral heads in 6-month-old Ctnnb1[Col2ER] mice. Red dotted lines: integral deformity. Red arrows: local collapse. Yellow arrows: subchondral bone destruction. (**C**) Quantitative analysis of BV/TV in the femoral heads in 6-month-old Ctnnb1[Col2ER] mice. (**D**) Quantitative analysis of subchondral bone defect area on the femoral head surface in older Ctnnb1[Col2ER] mice. (**E**) Alcian Blue Hematoxylin (ABH) staining of femoral heads in 6-month-old Ctnnb1[Col2ER] mice. (a–c) High-magnification images of representative subchondral bone region. Red dotted lines: integral deformity. Red arrows: local collapse. Black arrows: empty lacunae of osteoctyes. Black triangles: fat droplets. (**F**) Histomorphological quantitative analysis of empty lacunae rate. (**G**) Loading-bearing stiffness of the femoral heads in 6-month-old Ctnnb1[Col2ER] mice.

The online version of this article includes the following source data for figure 7:

**Source data 1.** Raw data for *Figure 7*.

subchondral bone destruction was identified in 6-month-old Ctnnb1[Col2ER] mice. Previous studies have revealed that strong trabeculae and integrated subchondral bone are critical to maintaining femoral head morphology (*Kawano et al., 2020*; *Motomura et al., 2011*). A significant decrease of loading-bearing stiffness confirmed a poor biomechanical support in 6-month-old Ctnnb1[Col2ER] mice. Therefore, it could be concluded that β-catenin inhibition-induced imbalanced osteogenic/adipogenic differentiation of Col2[+] cells caused a poor biomechanical support (manifesting with sparse trabeculae and subchondral bone destruction), eventually leading to femoral head collapse in Ctnnb1[Col2ER] mice.

Osteocytes are the bone cells with the longest life, up to decades within their mineralized environment (*Dallas et al., 2013*; *Bonewald, 2011*). However, numerous empty lacunae of osteocyte were observed in the necrotic femoral head of GONFH patients and glucocorticoid-induced rats. Glucocorticoids are known to induce cell apoptosis (*Weinstein, 2012*), and TUNEL staining confirmed increased osteocyte apoptosis in the human and rat necrotic femoral heads, leaving behind these empty lacunae. The importance of β-catenin in the regulation of apoptosis is well established in the skeletal system (*Jähn et al., 2012*; *Kitase et al., 2010*). Consistently, the expression of β-catenin was significantly decreased in the human and rat necrotic femoral heads. Moreover, we found that genetic deletion of *Ctnnb1* in Col2[+] cells spontaneously caused osteocyte apoptosis in mice and activation of β-catenin by systematically injecting Wnt agonist 1 alleviated osteocyte apoptosis in glucocorticoid-induced rats, which indicate that β-catenin inhibition-induced osteocyte apoptosis contributed to empty lacunae of GONFH.

There are some limitations to the present study. Firstly, Wnt agonist 1 is a cell-permeating activator of Wnt signaling that induces transcriptional activity dependent on β-catenin (*Lim et al.,*

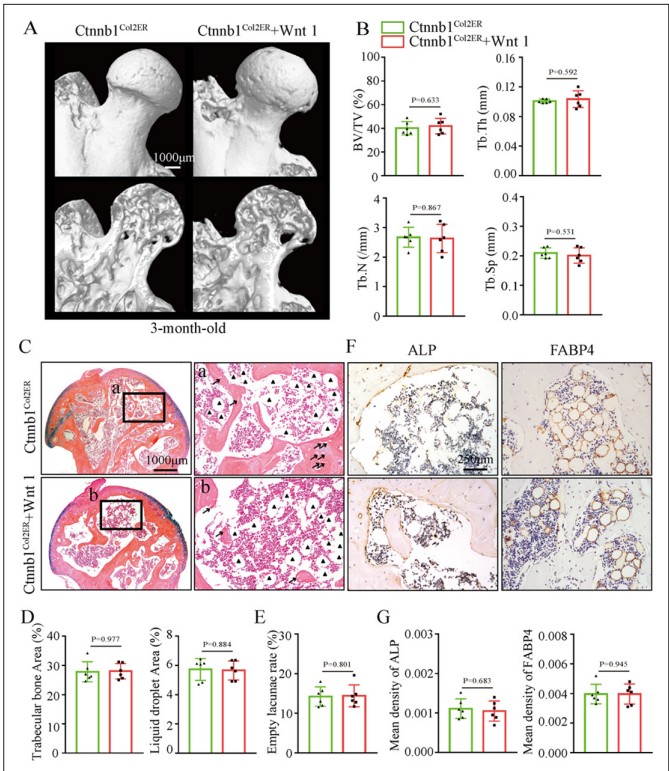

**Figure 8.** Systemic injection of Wnt agonist 1 cannot alleviate the glucocorticoid-induced osteonecrosis of the femoral head (GONFH)-like phenotype in Ctnnb1[Col2ER] mice. Ctnnb1[Col2ER] mice with five continuous dosages of tamoxifen (TM) injections (1 mg/10 g body weight) were treated with Wnt agonist 1 three times once a week until sacrifice at the age of 3 mo. (**A**) Representative μCT images of femoral heads in Wnt agonist 1-treated Ctnnb1[Col2ER] mice. (**B**) Quantitative analysis of BV/TV, Tb.N, Tb.Th, and Tb.Sp in the femoral heads of Wnt agonist 1-treated Ctnnb1[Col2ER] mice. (**C**) Alcian Blue Hematoxylin (ABH) staining of the femoral heads in Wnt agonist 1-treated Ctnnb1[Col2ER] mice. (a, b) High-magnification images of subchondral bone region. Black triangles: fat droplets. (**D**) Histomorphological quantitative analysis of trabecular bone area and fat droplet area in the femoral heads of Wnt agonist 1-treated Ctnnb1[Col2ER] mice. (**E**) Histomorphological quantitative analysis of empty lacunae rate in the femoral heads of Wnt agonist 1-treated Ctnnb1[Col2ER] mice. (**F, G**) Immunohistochemistry (IHC) staining and quantitative analysis of ALP and FABP4 in the femoral heads of Wnt agonist 1-treated Ctnnb1[Col2ER] mice.

The online version of this article includes the following source data for figure 8:

**Source data 1.** Raw data for *Figure 8*.

*2008*; *Kuncewitch et al., 2015*). Thus, Wnt agonist 1 alleviated rat GONFH by activating β-catenin signaling, but could not rescue GONFH-like phenotype in Ctnnb1[Col2ER] mice. Nevertheless, it remains unclear how Wnt agonist 1 works to activate β-catenin signaling. Secondly, Ctnnb1[Sp7ER] mice presented few lipid droplets and empty lacunae but a significant decrease in bone mass in the femoral heads. Previous studies revealed that specific knockout of *Ctnnb1* in Sp7-expressing cells promotes osteoclast formation and activity (*Yu et al., 2018*; *Song et al., 2022*), while its exact mechanism of bone mass loss still needs to be verified. Thirdly, that bone damage caused a poor mechanical support is the key to femoral head collapse. Although it is well known that an extensive subchondral bone destruction finally led to femoral head collapse in 6-month-old Ctnnb1[Col2ER] mice, the detailed mechanism of why this subchondral bone destruction occurs with age is still unclear.

In summary, our studies revealed that β-catenin inhibition induces the imbalance in osteogenic/adipogenic differentiation of BMSCs and plays a crucial role in GONFH pathogenesis. We also provided a *Col2*-specific *Ctnnb1* knockout mouse model for the first time, which mimics full spectrum of osteonecrosis phenotype of GONFH.

## Materials and methods

### Human specimen collection

Fifteen necrotic femoral head samples were collected from GONFH patients (ARCO grades III–IV) who received arthroplasty surgery. Ten non-necrotic femoral head samples were acquired from femoral neck fracture patients. Before the surgery, all patients signed the informed consent that their femoral head samples could be used for scientific research. All experiments with human specimens in the present study were approved by the Ethics Committee of the First Affiliated Hospital of Zhejiang Chinese Medical University (2018KL-005). The information of patients is listed in *Supplementary file 1*.

### Animals and interventions

Thirty 2-month-old male SD rats were purchased from the Animal Laboratory Center of Zhejiang Chinese Medical University (Zhejiang, China, SCXK 2014-0004) and randomly divided into sham group, GONFH group, and treatment group (n = 10 per group). We adopted Zheng's methods to establish a glucocorticoid-induced GONFH rat model (*Zheng et al., 2018*). Firstly, rats in the GONFH group and treatment group received one injection of lipopolysaccharide (0.2 mg/kg body weight, Sigma, USA) by tail vein, followed by three continuous intraperitoneal injections of 100 mg/kg body MPS (Pfizer, USA). In the subsequent 6 wk, MPS was intraperitoneally injected at the dosage of 40 mg/kg body weight three times every week. Rats in the treatment group were intravenously injected with Wnt agonist 1 (5 mg/kg body weight, Selleck, USA) at a synchronous frequency with MPS.

Col2a1-CreER$^{T2}$ mice, Ctnnb1$^{flox/flox}$ mice, Sp7-CreER$^{T2}$ mice, and Rosa26-LSL-tdTomato mice were obtained from Jackson Laboratory. To map cell fate of Col2$^+$ cells and Sp7$^+$ cells in the femoral head, Col2a1-CreER$^{T2}$;Rosa26-LSL-tdTomato (Tomato$^{Col2ER}$) mice and Sp7-CreER$^{T2}$;Rosa26-LSL-tdTomato (Tomato$^{Sp7ER}$) mice were induced by intraperitoneally injecting with tamoxifen (TM, 1 mg/10 g body weight/day, diluted in corn oil) for five consecutive days at the age of 2 or 4 wk, and traced as long as 9 months old. Mouse genotyping sequences are listed in *Supplementary file 2*. To further investigate the role of β-catenin in GONFH pathogenesis, Col2a1-CreER$^{T2}$;Ctnnb1$^{flox/flox}$ (Ctnnb1$^{Col2ER}$) mice and Sp7-CreER$^{T2}$;Ctnnb1$^{flox/flox}$ (Ctnnb1$^{Sp7ER}$) mice were generated and induced with five continuous injections of TM. After induction, Wnt agonist 1 (5 mg/kg body weight) was intravenously injected into Ctnnb1$^{Col2ER}$ mice three times every week until they were sacrificed at the age of 3 mo. All animal experiments were approved by the Animal Ethics Committee of Zhejiang Chinese Medical University (no. 20190401-10).

### Stereomicroscopy observation and biomechanical testing

Femoral head tissues harvested from human patients, glucocorticoid-induced GONFH rats, Ctnnb-1$^{Col2ER}$ mice, and Ctnnb1$^{Sp7ER}$ mice were viewed under a stereomicroscope (model C-DSD230, Nikon, Japan) to assess their appearance characteristics. The fresh femoral head tissues were placed on the test platform of a biomechanical testing machine (EnduraTec TestBench system, Minnetonka, MN). The axial compression load was applied at a speed of 0.5 mm/min until femoral head deformation. The loading-bearing stiffness of femoral head was calculated by testing software.

### μCT analysis

Femoral head samples were scanned with a micro-computed tomography (μCT, Skyscan 1176, Bruker μCT, Kontich, Belgium) at a resolution of 9 μm. Three-dimensional (3D) structure was reconstructed using NRecon Software v1.6 as described previously (*Xia et al., 2019*). The region of interest in the subchondral bone was selected for morphometric analysis, including bone volume fraction (BV/TV, %), average trabecular thickness (Tb.Th, mm), average trabecular number (Tb.N, 1/mm), average trabecular separation (Tb.Sp, mm) and subchondral bone defect area (%).

### Histology, immunohistochemistry, and histomorphometry

Femoral head tissues were processed into 3-μm-thick paraffin section or 10-μm-thick frozen section as described previously (*Xia et al., 2020a*). After deparaffinage and rehydration, sections were stained with ABH for 1 hr and then Orange G for 1 min for histological analysis. After ABH/Orange G staining, bone tissue presented yellow color and cartilage was blue. The trabecular bone area (%), fat droplet

area (%), and empty lacuna rate (%) were measured using OsteoMetrics software (Decatur, GA). The frozen sections were performed with 4',6-diamidino-2-phenylindole (DAPI) staining and hematoxylin and eosin (H&E) staining for evaluating *Cre*-recombination efficiency.

IHC assay was performed on the paraffin sections according to the previously established procedures (*Xia et al., 2020b*). Briefly, sections were immersed into 0.01 M citrate buffer (Solarbio, Beijing, China) at 60°C for 4 hr to repair antigen. After washing with phosphate buffer (PBS) three times, the sections were incubated with primary antibodies including β-catenin (diluted 1:500, abcam, ab32572, UK), Runx2 (diluted 1:200, Abcam, ab76956, UK), alkaline phosphatase (ALP, diluted 1:200, ARIGO, ARG57422, CN), osteocalcin (OCN; diluted 1:200, Takara, UK), PPAR-γ (diluted 1:300, Abcam,ab19481, UK), CEBP/α (diluted 1:400, Huabio, EM1710-02, CA), and fatty acid-binding protein (FABP4, diluted 1:200, abcam, ab92501, UK) overnight at 4°C. After incubation with secondary antibodies for 30 min at room temperature, tissue sections were stained with diaminobenzidine (DAB) solution to detect positive staining and hematoxylin for counterstaining. Mean density (IOD/area) as the parameter of immunohistochemical quantitative analysis was evaluated using Image-Pro Plus software (Media Cybernetics, Silver Spring, USA).

## Rat BMSC isolation and identification

BMSC isolation was conducted as previously described (*Zhu et al., 2020*). Briefly, bone marrow cells were isolated from the proximal femur of 4-week-old SD rats. After centrifugation at $200 \times g$ for 10 min, cells were resuspended with α-MEM (Gibco, Grand Island, NY) containing 10% fetal bovine serum (FBS, Gibco). By changing the medium 48 hr later, non-adherent cells were removed and the primary BMSCs were harvested. At 100% confluence, primary cultures were detached using trypsin and subcultured at $8 \times 10^3$ cells/cm$^2$. A total of 50 µL third passage (P3) BMSC suspension ($4 \times 10^6$/mL) were mixed with 5 µL of rat CD90-APC, rat anti-CD29-BV421, rat anti-CD45-FITC, or rat CD11b-PE (BD Pharmingen, Franklin Lakes, NJ) in each flow cytometry tube. After incubation at room temperature for 30 min followed by twice washes of a staining buffer, the cells were resuspended with 500 µL staining buffer for flow cytometry analysis.

## Rat BMSC intervention and staining

To investigate the role of glucocorticoids on BMSC differentiation, P3 BMSCs were treated with gradient concentrations of Dex (0, 10, $10^2$, $10^3$, $10^4$, $10^5$ nM) (Selleck, Shanghai, China) with or without SKL2001 (an agonist of β-catenin) (Selleck) for 72 hr. For oil red O staining, BMSCs were treated with adipogenic medium (α-MEM containing 10% FBS, 0.25 mM methylisobutylxanthine, 5 µg/mL insulin, and 50 µM indomethacin) for 72 hr followed by 5 µg/mL insulin alone for an additional 48 hr. After being fixed in 4% paraformaldehyde for 10 min, BMSCs were stained in 60% saturated oil red O solution for 5 min. For ALP staining, BMSCs were treated with osteogenic medium (α-MEM containing 10% FBS, 50 µg/mL ascorbic acid, and 5 mM β-glycerophosphate) for 14 d, fixed in 4% paraformaldehyde for 10 min, and stained with ALP staining kit (Thermo Fisher Scientific, Rockford, IL) for 15 min.

## Western blot analysis

According to the appearance and CT images, two pieces (about 2 * 2 * 2 cm$^3$) of necrotic tissues were cut from each human necrotic femoral head and the corresponding region from human fractured femoral head. These necrotic and non-necrotic tissues were mixed and ground into powder after being processed by liquid nitrogen, respectively. Tissue powder and rat P3 BMSCs were lysed by protein extraction reagent, and the liquid supernatant containing total protein was obtained through centrifuging at $12,000 \times g$ for 20 min. Then, proteins were separated by SDS-PAGE and transferred onto nitrocellulose membranes. The membranes were incubated overnight with primary antibodies including β-catenin (CST, 9582, USA), Runx2 (abcam, ab236639, UK), Osterix (huabio, ER1914-47, CA), ALP (arigo, ARG57422, CA), PPAR-γ (arigo, ARG55241, CA), and CEBP/α (CST, 8178s, USA) at a 1:1000 dilution. Next, the membranes were incubated with appropriate horseradish peroxidase-conjugated secondary antibodies at 37°C for 1 hr. The immune complexes were detected using a chemiluminescence kit and visualized via the ChemiDoc XRS Gel documentation system (Bio-Rad, CA). The gray intensity was measured using the software Quantity One (Bio-Rad).

## Statistical analysis

All data were presented as mean ± standard deviation (SD). Statistical analyses, including unpaired Student's $t$-test (two groups) and one-way ANOVA followed by Tukey's test (multiple groups), were performed with the SPSS 22.0 software. $*p<0.05$, $**p<0.01$ were considered statistically significant.

## Acknowledgements

This work was supported by the Chinese National Natural Science Foundation (grants nos. 82074469, 82104885, 82250710174 and 82161160342), Natural Science Foundation of Zhejiang Province (grants nos. LQ22H270001, LY21H270008, and LR23H270001), Ningbo medical key discipline (grant no. 2022-B01), and Ningbo Top Medical and Health Program (grant no. 2022020102).

## Additional information

### Funding

| Funder | Grant reference number | Author |
| --- | --- | --- |
| National Natural Science Foundation of China | 82104885 | Chenjie Xia |
| National Natural Science Foundation of China | 82074469 | Bangjian He |
| Natural Science Foundation of Zhejiang Province | LQ22H270001 | Chenjie Xia |
| Natural Science Foundation of Zhejiang Province | LY21H270008 | Bangjian He |
| Natural Science Foundation of Zhejiang Province | LR23H270001 | Hongting Jin |
| Ningbo medical key discipline | 2022-B01 | Chenjie Xia |
| Ningbo Top Medical and Health Program | 2022020102 | Chenjie Xia |
| National Natural Science Foundation of China | 82250710174 | Di Chen |
| National Natural Science Foundation of China | 82161160342 | Di Chen |

The funders had no role in study design, data collection and interpretation, or the decision to submit the work for publication.

### Author contributions

Chenjie Xia, Funding acquisition, Validation, Methodology, Writing – original draft, Writing – review and editing; Huihui Xu, Software, Validation, Investigation, Methodology; Liang Fang, Software, Validation, Methodology; Jiali Chen, Data curation, Software; Wenhua Yuan, Data curation, Formal analysis, Methodology; Danqing Fu, Visualization, Methodology; Xucheng Wang, Software, Methodology; Bangjian He, Resources, Software, Funding acquisition, Methodology; Luwei Xiao, Conceptualization, Resources, Funding acquisition; Chengliang Wu, Conceptualization, Resources, Data curation; Peijian Tong, Resources, Funding acquisition; Di Chen, Conceptualization, Validation, Writing – original draft, Project administration, Writing – review and editing; Pinger Wang, Conceptualization, Resources, Software, Supervision, Validation, Project administration; Hongting Jin, Conceptualization, Resources, Supervision, Funding acquisition, Validation, Writing – original draft, Project administration, Writing – review and editing

## Author ORCIDs

Chenjie Xia https://orcid.org/0000-0002-5987-4146
Di Chen https://orcid.org/0000-0002-4258-3457
Hongting Jin http://orcid.org/0000-0001-8795-0874

## Ethics

All experiments with human specimens were approved by the Ethics Committee of the First Affiliated Hospital of Zhejiang Chinese Medical University (2018KL-005). Before the surgery, all patients signed the informed consent that their femoral head samples could be used for scientific research.
All animal experiments were approved by the Animal Ethics Committee of Zhejiang Chinese Medical University (No. 20190401-10).

Reviewer #1 (Public Review): https://doi.org/10.7554/eLife.92469.3.sa1
Reviewer #2 (Public Review): https://doi.org/10.7554/eLife.92469.3.sa2
Reviewer #3 (Public Review): https://doi.org/10.7554/eLife.92469.3.sa3
Author Response https://doi.org/10.7554/eLife.92469.3.sa4

# Additional files

## Supplementary files

- Supplementary file 1. The information of patients who provided femoral head samples.
- Supplementary file 2. The primer sequences of mouse genotype.
- MDAR checklist

## Data availability

All data generated or analyzed during this study are included in the manuscript and supporting files.

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
