## [Editor Report · eLife assessment]

This study presents **valuable** findings on the mechanism of glucocorticoid-induced osteonecrosis of the femoral head. The data were collected and analyzed using **solid**, validated methodology and can be used as a starting point for functional studies of development of glucocorticoid-induced osteonecrosis. This article would be of interest to cell biologists and biophysicists working on potential pharmacological treatments for glucocorticoid-induced osteonecrosis.

---

## [Referee Report · Reviewer #1 (Public Review)]

Summary:

The manuscript by Xia et al. investigated the mechanisms underlying Glucocorticoid-induced osteonecrosis of the femoral head (GONFH). The authors observed that abnormal osteogenesis and adipogenesis is associated with decreased β-catenin in the necrotic femoral head of GONFH patients and inhibition of β-catenin signaling leads to abnormal osteogenesis and adipogenesis in GONFH rats. Of interest, deletion of β-catenin in Col2-expressing cells rather than in osx-expressing cells leads to a GONFH-like phenotype in femoral head of mice.

Strengths:

A strength of the study is that it sets up a Col2-expressing cell-specific β-catenin knockout mouse model that mimics full spectrum of osteonecrosis phenotype of GONFH. This is interesting and provides new insights into the understanding of GONFH. Overall, the data are solid and support their conclusions.

---

## [Referee Report · Reviewer #2 (Public Review)]

Summary:

In this manuscript, the authors reported a study to uncover that β-catenin inhibition disrupting the homeostasis of osteogenic/adipogenic differentiation contributes to the development of Glucocorticoid-induced osteonecrosis of the femoral head (GONFH). In this study, they first observed abnormal osteogenesis and adipogenesis associated with decreased β-catenin in the necrotic femoral head of GONFH patients, but the exact pathological mechanisms of GONFH remain unknown. They then performed in vivo and in vitro studies to further revealed that glucocorticoid exposure disrupted osteogenic/adipogenic differentiation bone marrow stromal cells (BMSCs) by inhibiting β-catenin signaling in glucocorticoid-induced GONFH rats, and specific deletion of β-catenin in Col2+ cells shifted BMSCs commitment from osteoblasts to adipocytes, leading to a full spectrum of disease phenotype of GONFH in adult mice.

Strengths:

This innovative study provides strong evidence supporting that β-catenin inhibition disrupts the homeostasis of osteogenic/adipogenic differentiation that contributes to the development of GONFH. This study also identifies an ideal genetic modified mouse model of GONFH. Overall, the experiment is logically designed, the figures are clear, and the data generated from humans and animals is abundant supporting their conclusions.

Weaknesses:

Lack of the discussion to explain how the Wnt agonist 1 works. There are several types of Wnt ligands. It is not clear if this agonist only targets Wnt1 or other Wnts as well? Also, why Wnt agonist 1 couldn't rescue the GONFH-like phenotype in β-cateninCol2ER mice needs to be discussed.

---

## [Referee Report · Reviewer #3 (Public Review)]

Summary:

In this manuscript, the authors are trying to delineate the mechanism underlying the osteonecrosis of the femoral head.

Strengths:

The authors provided compelling in vivo and in vitro data to demonstrate Col2+ cells and Osx+ cells were differentially expressed in the

the femoral head. Moreover, inducible knockout of β-catenin in Col2+ cells but not

Osx+ cells lead to a GONFH-like phenotype including fat accumulation, subchondral

bone destruction and femoral head collapse, indicating that imbalance of osteogenic/adipogenic differentiation of Col2+ cells play an important role in GONFH pathogenesis. Therefore, this manuscript provided the mechanistic insights of osteonecrosis as well as potential therapeutic target for disease treatment.

Weaknesses: Additional in depth discussion regarding the phenotype observed in mice is highly encouraged.

---

## [Author Response]

The following is the authors’ response to the original reviews.

**Reviewer #1 (Public Review):**
Summary:The manuscript by Xia et al. investigated the mechanisms underlying Glucocorticoid-induced osteonecrosis of the femoral head (GONFH). The authors observed that abnormal osteogenesis and adipogenesis are associated with decreased β-catenin in the necrotic femoral head of GONFH patients, and that the inhibition of β-catenin signalling leads to abnormal osteogenesis and adipogenesis in GONFH rats. Of interest, the deletion of β-catenin in Col2-expressing cells rather than in Osx-expressing cells leads to a GONFH-like phenotype in the femoral head of mice.Strengths:A strength of the study is that it sets up a Col2-expressing cell-specific β-catenin knockout mouse model that mimics the full spectrum of osteonecrosis phenotype of GONFH. This is interesting and provides new insights into the understanding of GONFH. Overall, the data are solid and support their conclusions.
**Reviewer #1 (Recommendations For The Authors):**
1. Fig. 1I should be quantified and presented as bar graphs to make it consistent with other data, and the significance should be shown.

Reply: Thanks for your comments. We have provided the quantitative bar graph in the new version.

1. Fig. 2H, beta-catenin, ALP and FABP4 should be labled below the X axis. Moreover, the pattern of Fig. 2H is different from other bar graphs and the dots for individual samples are missing, so I could not judge the N values for the experiments. N values should also be provided for Fig. 3.

Reply: Thanks for your comments. We have added the labels of beta-catenin, ALP and FABP4 below the X axis in Fig. 2H. The modes of quantitative bar graphs were changed to show the N values in the each experiment.

1. Fig. 4 shows the fate mapping of Col2+ cells and Osx+ cells in the femoral head. In this regard, the authors presented images for Col2-expressing cells at all the indicated time points, i.e. 1, 3, 6, and 9 months, but only presented images for Osx-expressing cells for 1 month while those for 3, 6, and 9 months are missing.

Reply: Thanks for your comments. Here, we showed that the expression of Osx+ cells in the femoral head were total different with Col2+ cells at the age of 3, 6 month, further indicating they were two different progenitor lineage cells.

**Author response image 1. sa4fig1:** 

1. Some experiments may need to be described in more detail" e.g., ABH/Orange G staining, biomechanical testing, μCT analysis, et al.

Reply: Thanks for your comments. We have provided more information of experiment procedures.

1. This study proposed that Col2-expressing cells play a key role in the progression of GONFH, did the authors use Col2+ cells for the in vitro experiments?

Reply: As in vitro experiments could not reflect the location of Col2-expressing cells in the femoral head, therefore here we applied in vivo lineage tracing study. After as long as 9 month of linage trace, we thoroughly showed the self-renew ability and osteogenic commitment of Col2+ cells, as well as its space variation in the femoral head with age. Conditional knockout of β-catenin caused that Col2+ cells trans-differentiated into adipogenic cells instead of osteogenic cells, which directly clarified the mechanism of Col2+ cells leading to GONFH-like phenotype in mice.

1. A few typo errors, such as Line 13, "contribute" should be "contributes"; Line 118, "reveled" should be "revealed".

Reply: We have revised the grammar errors in the new manuscript.

**Reviewer #2 (Public Review):**
Summary:In this manuscript, the authors reported a study to uncover that β-catenin inhibition disrupting the homeostasis of osteogenic/adipogenic differentiation contributes to the development of Glucocorticoid-induced osteonecrosis of the femoral head (GONFH). In this study, they first observed abnormal osteogenesis and adipogenesis associated with decreased β-catenin in the necrotic femoral head of GONFH patients, but the exact pathological mechanisms of GONFH remain unknown. They then performed in vivo and in vitro studies to further reveal that glucocorticoid exposure disrupted osteogenic/adipogenic differentiation of bone marrow stromal cells (BMSCs) by inhibiting β-catenin signaling in glucocorticoid-induced GONFH rats, and specific deletion of β-catenin in Col2+ cells shifted BMSCs commitment from osteoblasts to adipocytes, leading to a full spectrum of disease phenotype of GONFH in adult mice.Strengths:This innovative study provides strong evidence supporting that β-catenin inhibition disrupts the homeostasis of osteogenic/adipogenic differentiation that contributes to the development of GONFH. This study also identifies an ideal genetically modified mouse model of GONFH. Overall, the experiment is logically designed, the figures are clear, and the data generated from humans and animals is abundant supporting their conclusions.Weaknesses:There is a lack of discussion to explain how the Wnt agonist 1 works. There are several types of Wnt ligands. It is not clear if this agonist only targets Wnt1 or other Wnts as well. Also, why Wnt agonist 1 couldn't rescue the GONFH-like phenotype in β-cateninCol2ER mice needs to be discussed.

Reply: Thanks for your constructive comments. Wnt agonist 1 is a cell-permeating activator of the Wnt signaling pathway that induces transcriptional activity dependent on β-catenin (PMID: 25514428,18624906). In the present study, we aim to demonstrate that activation of β-catenin signaling could alleviate the phenotype of rat GONFH, thus only β-catenin and downstream targets (RUNX2, ALP, PPAR-γ, FABP4) expressions were detected after Wnt agonist 1 intervention. Conditional knockout β-catenin in Col2+ cells lead to an mouse GONFH-like phenotype. Wnt agonist 1 couldn't rescue this GONFH-like, as it did not activate β-catenin signaling. We have discussed them in the new version.

**Reviewer #3 (Public Review):**
Summary:In this manuscript, the authors are trying to delineate the mechanism underlying the osteonecrosis of the femoral head.Strengths:The authors provided compelling in vivo and in vitro data to demonstrate Col2+ cells and Osx+ cells were differentially expressed in the femoral head. Moreover, inducible knockout of β-catenin in Col2+ cells but not Osx+ cells lead to a GONFH-like phenotype including fat accumulation, subchondral bone destruction, and femoral head collapse, indicating that imbalance of osteogenic/adipogenic differentiation of Col2+ cells plays an important role in GONFH pathogenesis. Therefore, this manuscript provided mechanistic insights into osteonecrosis as well as potential therapeutic targets for disease treatment.Weaknesses:However, additional in-depth discussion regarding the phenotype observed in mice is highly encouraged.

Reply: Thanks for your comments. Inducible knockout of β-catenin in Col2+ cells but not Osx+ cells lead to a GONFH-like phenotype. Lineage tracing data showed Col2+ cells and Osx+ cells were different cell populations, and we have discussed the potential mechanism caused the different phenotypes between β-cateninCol2ER mice and β-cateninOsxER mice.

1. Why did the authors use dexamethasone in the cellular experiments but methylprednisolone to induce the GONFH rat model?

Reply: Thanks for the comments. Here, we applied a dexamethasone (DEX)-treated BMSC model in vitro and a methylprednisolone (MPS)-induced rat model in vivo for GONFH study based on the published literatures (PMID: 37317020, 29662787, 29512684,35126710, 32835568).

1. Both bone damage and fat accumulation were observed in 3-month-old and 6-month-old β-cateninCol2ER mice, but the femoral head collapse (the feature of GONFH at the late stage) only occurred in the older β-catenin Col2ER mice. This interesting observation needs to be discussed.Reply: Thanks for the comments. Bone damage caused a poor mechanical support is the key to femoral head collapse. Despite of similar trabecular bone loss and fat accumulation in the 3-month-old and 6-month-old β-cateninCol2ER mice, the older mice also presented extensive subchondral bone destruction. Integrated subchondral bone provided a well mechanical support for femoral head morphology, therefore femoral head collapse were occurred in the older β-cateninCol2ER mice.1. In the Materials and Methods, detailed information on the reagents should be provided.

Reply: We have provided detailed information of the important reagents.

1. As shown in Figure 4, β-cateninOsxER mice at 3 months of age did not show differences in lipid droplet area and empty lacunae rate, but there was a decrease in bone area. The authors should at least provide some necessary discussion of this phenomenon.

Reply: Thanks for your comments. In the present study, we found few lipid droplet and empty lacuna but a significant decrease of bone mass in the femoral heads of β-cateninOsxER mice. Previous studies showed that specific knockout of β-catenin in Osx-expressing cells promoted osteoclast formation and activity, leading to the bone mass loss (PMID: 29124436, 34973494). We discussed this phenomenon in the new version.